# Furoquinoline Alkaloids: Insights into Chemistry, Occurrence, and Biological Properties

**DOI:** 10.3390/ijms241612811

**Published:** 2023-08-15

**Authors:** Agnieszka Szewczyk, Filip Pęczek

**Affiliations:** 1Department of Pharmaceutical Botany, Faculty of Pharmacy, Jagiellonian University Medical College, Medyczna 9 Str., 30-688 Cracow, Poland; 2SSG of Medicinal Plants and Mushroom Biotechnology, Department of Pharmaceutical Botany, Jagiellonian University Medical College, Medyczna 9 Str., 30-688 Cracow, Poland; filip.peczek@student.uj.edu.pl

**Keywords:** furoquinoline alkaloids, bioactivity, Rutaceae, plant biotechnology

## Abstract

Furoquinoline alkaloids exhibit a diverse range of effects, making them potential candidates for medicinal applications. Several compounds within this group have demonstrated antimicrobial and antiprotozoal properties. Of great interest is their potential as acetylcholinesterase inhibitors and anti-inflammatory agents in neurodegenerative diseases. The promising biological properties of furoquinoline alkaloids have motivated extensive research in this field. As a result, new compounds have been isolated from this group of secondary metabolites, and numerous pharmacological studies have been conducted to investigate their activity. It is crucial to understand the mechanisms of action of furoquinoline alkaloids due to their potential toxicity. Further research is required to elucidate their mechanisms of action and metabolism. Additionally, the exploration of derivative compounds holds significant potential in enhancing their pharmacological benefits. In vitro plant cultures offer an alternative approach to obtaining alkaloids from plant material, presenting a promising avenue for future investigations.

## 1. Introduction

Furoquinoline alkaloids, derived from anthranilic acid, possess a furoquinoline backbone and are commonly found in the Rutaceae family. Notable alkaloids belonging to this group include dictamnine, skimmianine, kokusaginine, and γ-fagarine. Research has revealed that certain furoquinoline alkaloids exhibit pharmacological properties, including antibacterial, antifungal, antiviral, mutagenic, cytotoxic, antiplatelet, and enzyme-inhibitory effects [1]. The promising biological properties of furoquinoline alkaloids have driven extensive research into this compound group, leading to the discovery of new compounds and comprehensive pharmacological investigations. Given their potential toxicity, understanding the mechanisms of action of furoquinoline alkaloids is of the utmost importance. Additionally, exploring alternative methods, such as plant biotechnology, to obtain these metabolites has been an area of research interest. This study aims to provide a review of current research on furoquinoline alkaloids and prospects for future exploration. The review encompasses comprehensive searches of available databases such as Scopus, Google Scholar, Web of Science, MEDLINE-PubMed, BioMed Central, and Embase.

## 2. Biogenesis, Chemical Structure, and Occurrence of Furoquinoline Alkaloids

Furoquinoline alkaloids belong to the class of organic heterotricyclic compounds. Their distinctive chemical structure comprises the quinoline and furan rings. The quinoline ring is formed through a biosynthetic condensation process involving anthranilic acid and acetic acid, possibly in conjunction with one or both of the coenzyme-A derivatives. Alternatively, the condensation between anthranilic acid and a derivative of cinnamic acid can serve as a starting point, as demonstrated in tracer experiments with *Ruta angustifolia*. After the formation of the quinoline ring, in the case of furoquinoline alkaloids, furan rings are generated from isoprenoid-substituted compounds. Once 2,4-dihydroxyquinoline is formed, a dimethylallyl group attaches to the C-3 position. It is conceivable that the dimethylallyl group is initially added to the C-4 oxygen and subsequently migrates to C-3 [2].

Numerous recently published research papers have focused on the isolation of compounds from the furoquinoline alkaloid group. Apart from the well-known alkaloids, such as dictamnine (**1**), skimmianine (**2**), and γ-fagarine (**3**), several novel structures have been discovered. Table 1 presents examples of isolated compounds along with their corresponding plant sources and biological activity. Figure 1, Figure 2, Figure 3 and Figure 4 present the chemical structures of selected alkaloids. Most furoquinoline alkaloids are obtained from plants belonging to genera such as *Ruta* sp., *Dictamnus* sp., *Zanthoxylum* sp., *Haplophyllum* sp., and *Euodia* sp. These plants primarily belong to the Rutaceae family [1]. However, there are also reports of these alkaloids occurring in other families, such as in Apiaceae. For instance, two furoquinoline-type alkaloids were isolated from *Ammi majus* (4-hydro-7-hydroxy-8-methoxyfuroquinoline (**4**) and 4-hydro-7-hydroxy-8-prenyloxyfuroquinoline (**5**)) [3].

## 3. Pharmacological Properties

Furoquinoline alkaloids demonstrate multidirectional pharmacological activity. These compounds exhibit antimicrobial, antiprotozoal, antispasmodic, and anti-inflammatory effects. They also possess antiosteoporosis properties and antiplatelet aggregation effects. Additionally, they can influence enzymatic activity, such as the potent antiacetylcholinesterase activity observed in skimmianine (**2**). Furthermore, these alkaloids often exhibit cytotoxic effects, making them potentially useful in the treatment of certain cancers. However, it is important to note that they can also have mutagenic properties. A well-known example is the alkaloid dictamnine (**1**), which has been documented to have mutagenic and genotoxic effects.

### 3.1. Antibacterial Activity

Furoquinoline alkaloids frequently exhibit antibacterial activity. Skimmianine (**2**), γ-fagarine (**3**), and isohaplopine (**7**) have demonstrated activity against seven Gram(+) bacteria, *Bacillus subtilis*, *B. cereus*, *Staphylococcus aureus*, *S. epidermidis*, *Streptococcus pyogenes*, *Enterobacter aerogenes*, and *Enterococcus* sp., and eight Gram(−) bacteria, *Escherichia coli*, *Klebsiella pneumonia*, *Pseudomonas aeruginosa*, *Enterobacter cloacae*, *Shigella sonnei*, *Salmonella typhimurium*, *Burkholderia cepacia*, and *Morganella morganii* [1]. Flindersiamine (**35**) isolated from *E. yaaxhokob* exhibited moderate antimicrobial activity against *S. aureus* and *S. faecalis*. Further alkaloids such as kokusaginine (**8**), skimmianine (**2**), and haplopine (**6**) exhibited photo-activated antimicrobial activity against *S. aureus*. Pteleine (**9**) possessed moderate antimicrobial activity against *M. smegmatis*, *B. subtilis*, and *S. aureus* (MIC 4.39–87.8 μM) [20]. Furoquinoline alkaloids isolated from *Vepris lecomteana* were tested for antibacterial activity against bacteria such as *E. coli*, *B. subtilis*, *Pseudomonas agarici*, *Micrococcus luteus*, and *Staphylococcus warneri*. The most active was lecomtequinoline A (**10**). However, crude extracts exhibited higher antibacterial activity, indicating a potential synergistic effect of its metabolites [19]. Megistoquinone I (**47**), isolated from *S. megistophylla*, showed antibacterial properties against two Gram-positive, *S. aureus* and *S. epidermidis* (MIC 9.073 and 10.7mM, respectively), and four Gram-negative bacteria, *Pseudomonas aeruginosa*, *E. coli*, *Enterobacter cloacae*, and *Klebsiella pneumoniae* (MIC 12.5, 18.3, 12.0, 20.3 mM, respectively) [20]. Dictamnine (**1**), robustine (**11**), and γ-fagarine (**3**) isolated from the root bark of *Dictamnus dasycarpus* displayed moderate inhibitory activity against *Bacillus subtilis* and *Pseudomonas aeruginosa*, with MICs of 32–64 μg/mL [11]. Both (+)- and (−)-zanthonitidine A (**48**) demonstrated moderate inhibitory activity against *Enterococcus faecalis* and *Staphylococcus aureus*, with MIC values of 21.97, 21.97 µg/mL and 12.54, 25.09 µg/mL, respectively. Robustine (**11**) showed inhibitory activity against *Enterococcus faecalis*, with MIC values of 5.37 µg/mL [33]. Alkaloids such as γ-fagarine (**3**), maculine (**36**), and flindersiamine (**35**) from the bark of *Helietta apiculata* demonstrated inhibitory activity against *Bacillus cereus*, with MICs ranging from 6.2 to 12.5 μg/mL [34]. It should be noted that antimicrobial activity may be associated with a phototoxic reaction. For instance, maculine (**36**) showed phototoxic activity against *Streptococcus faecalis* in long-wave UV light [35].

### 3.2. Antiviral Activity

The quest for novel, effective antiviral drugs has become increasingly crucial in recent years. Furoquinoline alkaloids have also been studied for their potential as antivirals. γ-Fagarine (**3**) and kokusaginine (**8**), isolated from Ruta angustifolia, exhibited antiviral properties against HCV, with IC_50_ values of 20.4 and 6.4 μg/mL, respectively [1]. Three furoquinoline alkaloids, γ-fagarine (**3**), haplopine (**6**), and (+)-platydesmine (**49**), exhibited antiviral activity against HIV-1 at low concentrations (EC_50_ < 5.85 μM). The best therapeutic index was shown for γ-fagarine [20]. Kokusaginine (**8**) demonstrated moderate anti-HCV (hepatitis C virus) activity [36]. Skimmianine (**2**) and dictamnine (**1**) function as DNA-intercalating agents, thereby inhibiting viral replication within the host cells. These compounds have been singled out, alongside other quinoline alkaloids, as promising compounds in combatting SARS-CoV-2 [37,38,39].

### 3.3. Antifungal Activity

Certain alkaloids possess antifungal properties that are relevant to both human pathogens and plant crop pathogens. Skimmianine (**2**), γ-fagarine (**3**), and isohaplopine (**7**) demonstrated activity against *Candida albicans*, *C. tropicali*, *C. krusei*, *C. parapsilosis*, *Saccharomyces cerevisiae*, *Cryptococcus neoformans*, and *C. gattii* [1]. Flindersiamine (**35**), kokusaginine (**8**), skimmianine (**2**), dictamnine (**1**), maculine (**36**), and platydesmine (**49**) showed inhibition of the growth of the fungus *L. gongylophorus*, a symbiotic fungus of the insect pest *Atta sexdens rubropilosa* [20]. Dictamnine (**1**), robustine (**11**), and γ-fagarine (**3**) isolated from the root bark of *Dictamnus dasycarpus* exhibited moderate inhibitory activity against *Candida albicans* [11]. Maculine (**36**) exhibited phototoxic activity against *Saccharomyces cerevisiae* and *Candida albicans* under long-wave UV light [35]. Alkaloids such as dictamnine (**1**), γ-fagarine (**3**), maculine (**36**), flindersiamine (**35**), and kokusaginine (**8**) from the bark of *Helietta apiculata* demonstrated inhibitory activity against *Candida krusei*, with MICs ranging from 25 to 50 μg/mL [34]. Furoquinoline alkaloids found in *Zanthoxylum armatum* displayed activity against *Aspergillus flavus* and *Aspergillus niger*, although individual compounds were not studied [40]. Dictamnin (**1**), isolated from *Dictamnus dasycarpus*, exhibited antifungal activity against the plant pathogen *Cladosporium cucumerinum* (MIC 125.628 μM) [1]. The furoquinoline alkaloid 5-(1,1-dimethylallyl)-8-hydroxy-furo[2-3-b]quinoline (**12**), isolated from *Ruta chalepensis*, demonstrated antifungal properties against *Rhizoctonia solani*, *Sclerotium rolfsai*, and *Fusarium solani*, which are responsible for root rot and wilt disease in potatoes, tomatoes, and sugar beets. Skimmianine (**2**), maculine (**36**), kokusaginine (**8**), and flindersiamine (**35**), isolated from *Raualinoa echinata*, exhibited antifungal properties against another plant pathogen, *Leucoagaricus gongylophorus* [1].

### 3.4. Antifouling Activity

The exploration of novel, environmentally friendly antifouling agents has revealed the potential of certain furoquinoline alkaloids. Kokusaginine (**8**) and flindersiamine (**35**) derived from *Balfourodendron riedelianum* demonstrated notable antifouling activity in tests conducted with mussel *Mytilus edulis platensis* [41].

### 3.5. Antiprotozoal and Anti-Insect Activity

Malaria is a highly prevalent disease worldwide, with an annual incidence of more than 200 million cases and a mortality rate ranging from 1 to 3 million deaths. It is caused by several species of *Plasmodium* protozoa. Extensive research is ongoing to prevent and treat this disease [42,43]. Furoquinoline alkaloids have long been recognized for their antiprotozoal activity, particularly against pathogens of the genera *Plasmodium*, *Leishmania*, and *Trypanosoma*. In vitro studies have confirmed the antiprotozoal activity of alkaloids such as skimmianine (**2**), haplopine (**6**), kokusaginine (**8**), acronydine (**50**), and acronycidine (**13**) against *Plasmodium falciparum*, *Leishmania donovani*, *Trypanosoma cruzi*, and *T. brucei*. Among these, skimmianine exhibited the highest antiprotozoal activity against *T. cruzi* and *T*. *brucei* [1]. Five furoquinoline alkaloids tested, kokusaginine (**8**), skimmianine (**2**), haplopine (**6**), acronycidine (**13**), and acronydine (**50**), showed antiprotozoal activity against HB3 (chloroquine-sensitive) and W2 (chloroquine-resistant) clones of *P. falciparum*. The most active compound was found to be acronydine (**18**), (IC_50_ 22.6 and 4.63 μM, respectively) [20]. The prevention of malaria also involves targeting mosquito larvae. Alkaloids found in *Ruta chalepensis*, particularly 5-(1,1-dimethylallyl)-8-hydroxyfuro[2-3-b]quinolone (**12**), showed efficacy against *Culex pipiens* larvae, while posing minimal or no risk to humans and animals [1]. Another alkaloid demonstrating activity (IC_50_ = 35 µM) against *Plasmodium falciparum* is heliparvifoline (**16**), isolated from *Melicope madagascariensis* [21]. Two furoquinoline alkaloids, isolated from *Melicope moluccana*, leptanoine C (**15**) and haplopine-3,3′-dimethylallyl ether (**17**), demonstrated antimalarial activity against *Plasmodium falciparum* 3D7, with IC_50_ values of 0.18 ppm and 2.28 µg/mL, respectively [22]. Furoquinoline alkaloids (pteleine (**9**), maculine (**36**), skimmianine (**2**), robustine (**11**), γ-fagarine (**3**), and dictamnine (**1**)) found in the roots of *Zanthoxylum tingoassuiba* exhibited antiprotozoal activity against *Leishmania amazonensis* and *Trypanosoma cruzi*, comparable with positive controls (benzonidazole and amphotericin B) [44]. Skimmianine, (**2**) at a concentration above 37.5 µg/mL, displayed an inhibitory effect on *Trichomonas vaginalis* trophozoites [45].

### 3.6. Anticancer and Cytotoxic Activity

A large number of furoquinoline alkaloids have a significant cytotoxic effect. However, some isolated furoquinoline alkaloids, such as confusadine (**14**), furomegistines I (**51**) and II (**52**), and megistosarconine (**53**), exhibit only weak or no cytotoxic activity on different tumor cell lines [20]. Dictamnine (**1**) exhibits anticancer activity in non-small-cell lung cancer, breast cancer, and CNS cancer. However, due to its toxicity, it is more reasonable to consider using derivatives of this alkaloid [1]. Dictamnine (**1**) isolated from *Ruta graveolens* showed greater cytotoxic activity against HeLa (EC_50_ 12.6 μM) compared to the KB tumor cell lines (EC_50_ 103 μM) [20]. Three furoquinoline alkaloids isolated from the genus *Haplophyllum*, namely γ-fagarine (**3**), skimmianine (**2**), and haplopine (**6**), showed cytotoxic activity against the HeLa cell line (IC_50_ <50.0 μM) [20]. Several furoquinoline alkaloids (skimmianine (**2**), evolitrine (**18**), kokusaginine (**8**), and maculosidine (**19**)) isolated from *Acronychia laurifolia* have shown different cytotoxic potencies against a specific human cancer cell line, BC1 (EC_50_ 15.4, 25, 3, >70, and >70 μM), KB-V1^+^ (17.0, 12.7, 17.0, and 17.4 μM, respectively) and the KB-V1^−^ cell line (10.8, 16.6, 55, 6, and 39.4 μM, respectively) [20]. Dictamnine (**1**), γ-fagarine (**3**), and skimmianine (**2**) were identified as moderate cytotoxic agents against murine leukemia cell lines P-388, A549, and HT-29 [20]. Meanwhile, 7-(2′-hydroxy-3′-chloroprenyloxy)-4-methoxyfuroquinoline (**20**), 7-(2′,3′-epoxyprenyloxy)-4-methoxyfuroquinoline (**54**), pteleine (**9**), and (+)-7 8-dimethoxymyrtopsin (**55**) were isolated from the genus *Melicope*. Compounds **20** and **54** exhibited cytotoxic activity against the HeLa cell line (IC_50_ 34 and 20.1 μM, respectively). Compound **55** showed activity against the P-388 cell line (EC_50_ 39.0 μM). Medicosmine (**56**) isolated from *Boronella konambiensis* showed a slight cytotoxic effect against the murine leukemia cell line L1210 (IC_50_ 48.0 μM) [20]. Skimmianine (**2**) demonstrated cytotoxicity against various cancer cell lines and genotoxicity. It suppressed the proliferation and migration of human esophageal squamous cell carcinoma by blocking the activation of ERK1/2 [46]. Isodictamnine (**39**), iso-γ-fagarine (**40**), and γ-fagarine (**3**), occurring in *Glycosmis arborea*, showed inhibitory effects toward the tumor promoter 12-O-tetradecanoylphorbol 13-acetate-induced Epstein–Barr virus early antigen [20]. Compounds isolated from *Zanthoxylum buesgenii*, including maculine (**36**) and kokusaginine (**8**), demonstrated greater activity than doxorubicin on several tested cancer cell lines (leukemia CCRF-CEM and CEM/ADR5000, breast cancer MDA-MB231 and its resistant sub-line MDAMB231/BCRP, colon cancer HCT116p53 and its resistant sub-line, glioblastoma U87MG, hepatocarcinoma ∆EGFR, and HepG2) [1]. Maculine (**36**) and its derivatives, 5-methoxymaculine (**37**) and 5,8-dimethoxymaculine (**38**), as well as 4,5,6,7,8-pentamethoxyfuroquinoline (**21**) and flindersiamine (**35**), found in *Vepris punctate*, showed weak cytotoxic activity against the A2780 cell line (IC_50_ < 20 μM) [20]. Furoquinoline alkaloids, such as 5-methoxyrobustine (**22**), dictamnine (**1**), robustine (**11**), isopteleine (**41**), and γ-fagarine (**3**), presented weak cytotoxic activity against MCF-7 cells. Two furoquinoline alkaloids isolated from *Aegle marmelos*, namely aegelbine A (**23**) and B (**24**), were weakly active against the MCF-7 as well as HepG2 and PC-3 cell lines. Moreover, 7-isopentenyloxy-γ-fagarine (**25**) exhibited significant cytotoxicity against the Raji (lymphoma) and Jurkat (leukemia) cell lines (IC50 values of 1.5 and 3.6 μg/mL, respectively). Compound **25** possessed also cytotoxic activity against the MCF-7 cell line (IC50 = 15.5 μg/mL). Meanwhile, 7-isopentenyloxy-γ-fagarine (**25**), skimmianine (**2**), and perfamine (**57**) showed moderate to low cytotoxicity against KG-1a (leukemia) and HEp-2 (HeLa contaminant) and higher activity against the multidrug-resistant HL-60/MX1 cell line, compared with the control etoposide (*p* < 0.05) [26]. Skimmianine (**2**) demonstrated moderate cytotoxicity (IC_50_ value of 1.5 µM) against the colon cancer cell line HT-29 [21]. Confusameline (**26**), isolated from the leaves of *Melicope semecarpifolia*, exhibited cytotoxic activity against the P-388, HT-29, and A549 cell lines in vitro, with the highest cytotoxicity (ED_50_ value = 0.03 μg/mL) observed against the P-388 cell line [27]. Dictamnine (**1**) inhibited lung cancer growth in vitro and in vivo by downregulating the PI3K/AKT/mTOR and MAPK signaling pathways. Its mechanism of action is based on direct binding to c-Met and the inhibition of its phosphorylation. The synergistic effect of dictamnine and gefitinib or osimertinib on EGFR-TKI-resistant lung cancer cells is noteworthy. Dictamnine shows potential as a therapeutic agent for lung cancer or other tumors with an overactive c-Met pathway [47]. Skimmianine (**2**), isolated from *Zanthoxylum leprieurii*, demonstrated cytotoxicity in the HeLa cell line, with an IC_50_ value of 12.8 µg/mL [48]. Two furoquinoline alkaloids, 4-hydro-7-hydroxy-8-methoxyfuroquinoline (**4**) and 4-hydro-7-hydroxy-8-prenyloxyfuroquinoline (**5**), isolated from *Ammi majus*, exhibited cytotoxic and antiproliferative activity against HepG-2 and MCF-7 (IC_50_ = 230.2 and 326.5 μM) and against MCF-7 (IC 50 = 234.2 μM), respectively [3].

### 3.7. Anti-Inflammatory and Antioxidant Activity

Evolitrine (**18**) effectively inhibited (57% inhibition at the dose of 20 mg/kg) the formation of edema caused by subplantar carrageenan injection in rats but did not cause side effects [20]. Seven furoquinoline alkaloids isolated from *Dictamnus dasycarpus*, namely skimmianine (**2**), dictamnine (**1**), γ-fagarine (**3**), iso-γ-fagarine (**40**), haplopine (**6**), dictangustine-A (**42**), and isomaculosidine (**43**), showed anti-inflammatory activity. Skimmianine (**2**) showed the most potent inhibition (IC50 values of 7.0 μM) of LPS-induced NO production in BV2 cells. Alkaloid glycoside clausenaside F (**27**) obtained from *Clausena lansium* exhibited moderate inhibitory effects on LPS-induced NO production in BV2 cells [26]. Recently, skimmianine (**2**) has demonstrated notable anti-inflammatory effects in ear and paw edema models. The alkaloid’s mechanism of action is multifaceted and includes the inhibition of the transcription of the TNF-α and IL-6 genes; the inhibition of the production of nitric oxide, prostaglandin E2, and superoxide anions; and the release of elastase [49,50]. Haplopine-3,3′-dimethylallyl ether (**17**), isolated from *Vepris glomerata*, showed a strong antioxidant effect. The alkaloid inhibited superoxide anion production (IC_50_ = 13.0 μM) and elastase release (IC_50_ = 19.6 μM). Dictamnine (**1**) and robustine (**11**) showed weaker antioxidant activity [26].

### 3.8. Anticholinesterase and Neuroprotective Activity

Skimmianine (**2**) shows potential in the treatment of Alzheimer’s disease due to its ability to inhibit acetylcholine esterase (AChE) and the production of nitric oxide (NO), which can influence inflammation within the nervous tissue [1]. The neuroprotective effect of skimmianine may also be attributed to its ability to inhibit neuroinflammation in LPS-activated microglia by targeting the NF-κB activation pathway [49]. Furoquinoline alkaloids extracted from the leaves of Evodia lepta were evaluated for their cholinesterase (ChEs)-inhibitory activity. Kokusaginine (**8**) and melineurine (**28**) exhibited the highest activity against AChE and BChE, respectively [51]. Evolitrine (**18**) significantly affected neurite growth mediated by nerve growth factor in PC12 cells (EC50 3.8 μg/mL), and thus it may have potential application in the treatment of Alzheimer’s disease [26].

### 3.9. Cardiovascular and Antiplatelet Activity

Furoquinoline alkaloids also can show cardiovascular effects. Robustine (**11**), confusameline (**26**), γ-fagarine (**3**), skimmianine (**2**), haplopine (**6**), evolitrine (**18**), kokusaginine (**8**), and dictamnine (**1**) inhibited human phosphodiesterase 5, which is responsible for regulating intracellular cGMP levels and vascular smooth muscle [20]. Skimmianine (**2**) affected vasopressor responses in an animal model in rats [52]. Dictamnine (**1**) at a concentration of 100 μg/mL inhibited the platelet aggregation induced by arachidonic acid, collagen, and PAF. Furoquinoline alkaloids such as γ-fagarine (**3**), skimmianine (**2**), haplopine (**6**), and robustine (**11**) also showed the inhibition of aggregation induced by thrombin, arachidonic acid, collagen, and PAF in tested rabbit platelets. Significant inhibitory effects on platelet aggregation also were observed for confusameline (**26**), confusadine (**14**), evolitrine (**18**), kokusaginine (**8**), and pteleine (**9**) [20]. Furoquinoline alkaloids 8-hydroxy-9-methyl-furo[2,3-b]quinolin-4(9H)-one (**44**), isopteleine (**41**), robustine (**11**), γ-fagarine (**3**), isodictamnine (**39**), and skimmianine (**2**), isolated from *D. angustifolius*, showed significant antiplatelet effects [26].

### 3.10. Antiosteoporosis Activity

Dictamnine (**1**) has emerged as a promising therapeutic agent for the treatment of osteoporosis. This compound effectively mitigates osteoclast formation and demonstrates efficacy in alleviating OVX-induced osteoporosis [53].

### 3.11. Antinociceptive Action

Choisyine (**58**) exhibits antinociceptive activity in animal models. Studies have revealed that the cholinergic pathway plays a role in mediating its antinociceptive effects [31,54].

### 3.12. 5-HT2 Receptor Inhibition

Furoquinoline alkaloids such as skimmianine (**2**), kokusaginine (**8**), and confusameline (**26**) have been found to act on 5-HT receptors, particularly the 5-HT2 subtype. Among these alkaloids, skimmianine exhibits the most potent inhibitory effect [1].

### 3.13. Anti-Anaphylactoid Activity

Dictamnine (**1**) demonstrated anti-anaphylactoid activity in a mouse model of hind paw extravasation. It is suggested that dictamnine may act through the MrgX2 receptor located on mast cells, making it a promising candidate for an effective anti-anaphylactic compound [55].

### 3.14. Estrogenic Activity

Three alkaloids, γ-fagarine (**3**), skimmianine (**2**), and haplopine (**6**), in doses of 10 mg/kg, showed estrogenic activity in immature rats. The uterine weight increased by 193.9%, 22.6%, and 74.4%, respectively [20].

### 3.15. Hepatoprotective Activity

Hepatoprotective effects against DL-galactosamine-induced damage in WB-F344 cells and inhibitory effects on the LPS-induced NO production of the compounds found in *Clausena emarginata* were investigated. γ-Fagarine (**3**) showed hepatoprotective effects against DL-galactosamine-induced toxicity [15]. 

## 4. Discussion

Natural compounds are a promising group that can be used to search for new, effective drugs in the treatment of many diseases. Furoquinoline alkaloids belong to a very large group of quinoline alkaloids and attract the attention of many researchers. The development of laboratory techniques, including methods of metabolite isolation, means that new compounds are still being isolated. The basic structure of this group of alkaloids consists of a furoquinoline ring. The compounds occur in the form of aglycones; however, there has also been a report of a glycoside form of the alkaloid, namely clausenaside F (**27**) [26]. An interesting form comprises dimeric combinations of furoquinoline alkaloids such as anstifolines A (**59**) and B (**60**) [32].

For many years, plants used in folk herbal medicine have been intensively studied. They are often endemic and very rare species. This research aimed, among other goals, at examining the chemical composition and medicinal properties of plants. Biological research most often concerns extracts from various parts of plants, and the individual compounds that are part of the extract are less frequently studied. There is still much to explore in this field. Of the furoquinoline alkaloids isolated so far, only some have been analyzed for their biological activity. Well-documented healing properties are only noted for the most common alkaloids, such as dictamin, skimmianine, or γ-fagarine. Less common alkaloids, or alkaloids occurring in small amounts, do not have a sufficiently documented biological effect [26].

Furoquinoline alkaloids are a group of compounds found mainly in the Rutaceae family. However, there are reports of the occurrence of this type of alkaloid also in another family known for its medicinal properties—Apiaceae. This creates new possibilities of exploitation for researchers [3]. 

Discovering new molecular structures is extremely important in the search for new drugs. The molecular structures of natural compounds can provide valuable clues for drug design. A very good example of such an application is the discovery of febrifugin, triptanthrin, and their analogues, with significant antimalarial activity. New classes of compounds have allowed the huge expansion of research into malaria therapy [26].

Furoquinoline alkaloids exhibit diverse effects, making them potential candidates for medicinal use. Many compounds in this group demonstrate antimicrobial and antiprotozoal activity [23,56]. The use of furoquinoline alkaloids as inhibitors of AChE and anti-inflammatory agents in neurodegenerative diseases is particularly interesting [26]. Furoquinoline alkaloids show great potential for cytotoxic and anticancer activity [57,58,59]. However, it is important to note that alkaloids can also have toxic effects. Studies on the toxicity of individual alkaloids are scarce and concern only a few selected compounds. Dictamnine, in particular, is well known for its phototoxic and photomutagenic properties [60]. γ-Fagarine induced sister chromatid exchange in human lymphocytes. γ-Fagarine, skimianine, and dictamnine showed strong mutagenic activity in *S. typhimurium* strains TA98 and TA100, but showed relatively little or no activity in the corresponding R-factor-free strains TA1538 and TA1535. Furoquinoline alkaloids can be activated as mutagenic metabolites by cytochrome P450 and cytochrome P448 and possibly a flavin-containing monooxygenase [20].

To address this, exploring the synthesis of derivatives from native alkaloids with improved therapeutic profiles and reduced toxicity is an interesting approach. The modification of dictamnine allowed fluorinated analogues with improved cardiac effects to be obtained. The mechanism of their action might be related to epinephrine α receptors, the M-receptor, and the calcium channel receptor [61]. Derivatives of cocusaginine and flindersiamine, the primary alkaloids isolated from the bark of *Balfourodendron riedelianum*, have shown superior in vitro activity against *Trypanosoma cruzi* compared with positive controls [62]. 

As climate change and diminishing crop areas pose challenges in obtaining plant metabolites from field crops, plant biotechnology offers an alternative solution. In vitro plant cultures provide a means to produce valuable plant metabolites independent of environmental conditions and seasonal constraints. Furoquinoline alkaloids can be produced in vitro. Species of the genus *Ruta* have been found to contain alkaloids such as γ-fagarine and skimmianine in culture [63,64]. The use of innovative biotechnological technologies enables the increased production of secondary metabolites and biomass through the utilization of temporary immersion bioreactors. In in vitro cultures of *Ruta montana* carried out in Plantform^TM^ bioreactors, the highest content of γ-fagarine and skimmianine was 305.4 mg/100 g DM and 233.7 mg/100 g DM, respectively [65]. In turn, in in vitro cultures of *Ruta chalepensis* carried out in RITA^®^ bioreactors, the highest content of γ-fagarine and skimmianine was 186.8 mg/100 g DM and 291.6 mg/100 g DM, respectively [66].

## 5. Conclusions

Furoquinoline alkaloids exhibit a diverse range of therapeutic properties, such as antimicrobial, antiprotozoal, and anti-inflammatory effects. Furthermore, their ability to inhibit acetylcholinesterase makes them potential candidates for the treatment of neurodegenerative diseases, such as Alzheimer’s disease. 

Although some mechanisms of action of these compounds are well known, there is still a considerable amount of research needed to understand their mechanisms of action and metabolism. Additionally, exploring the development of derivatives holds significant promise in enhancing their pharmacological effects. The discussed group of compounds may contribute to the development of new drugs in the future through targeted pharmacological modeling and synthetic modification.

To source alkaloids from plant material, in vitro plant cultures offer an alternative to traditional field crops and can serve as a valuable method for alkaloid production. Obtaining natural compounds from plants can be associated with many problems related to both field cultivation and harvesting from natural sites. An alternative may be the use of in vitro plant cultures. Research on in vitro cultures should be aimed at maximizing the production of selected secondary metabolites. Plant biotechnology offers many opportunities in this field. In addition to manipulating the composition of the medium, different lighting conditions, elicitation with biotic and abiotic elicitors, as well as the addition of precursors of metabolic pathways can be used. This approach opens up opportunities for the efficient and controlled production of these compounds, paving the way for further studies and applications in various fields of research.

## Figures and Tables

**Figure 1 ijms-24-12811-f001:**
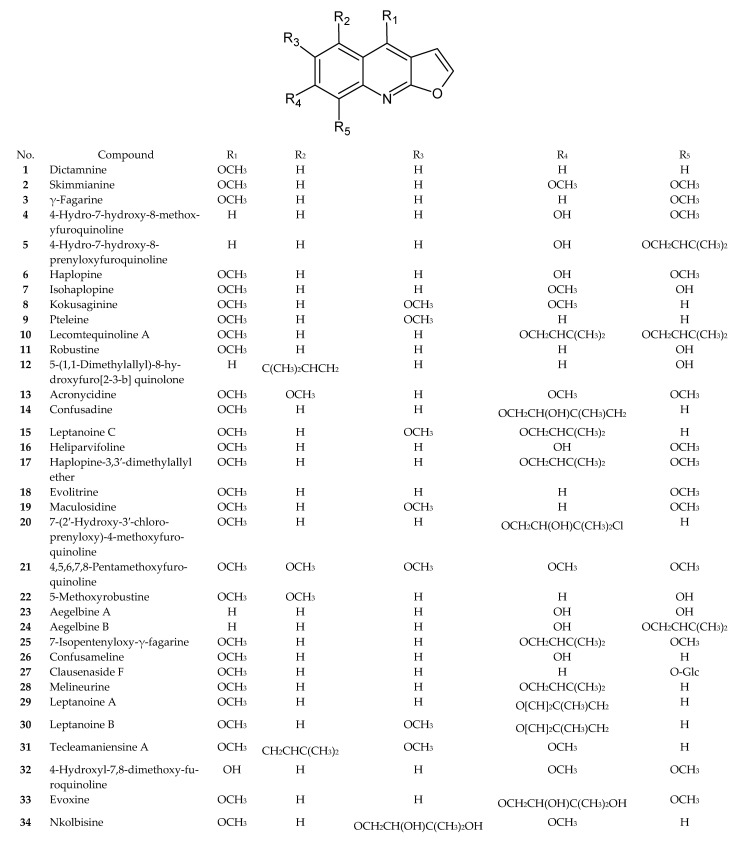
Chemical structures of compounds **1**–**34**.

**Figure 2 ijms-24-12811-f002:**
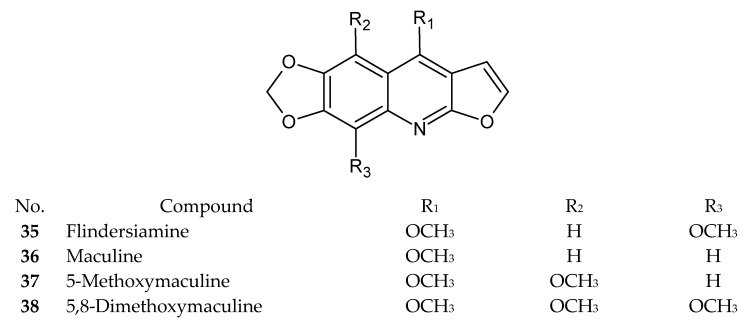
Chemical structures of compounds **35**–**38**.

**Figure 3 ijms-24-12811-f003:**
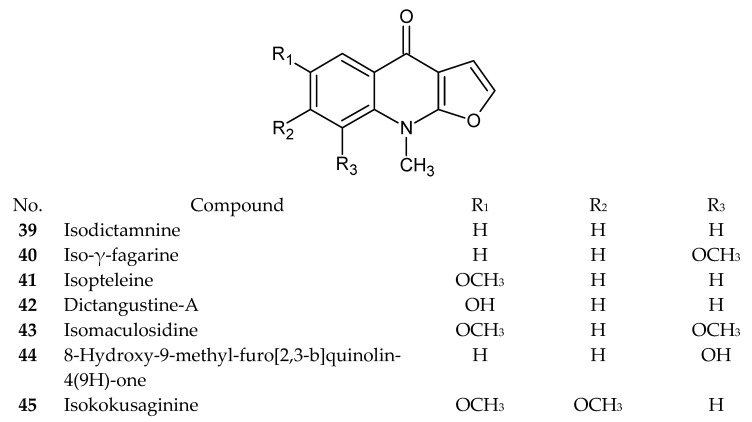
Chemical structures of compounds **39**–**45**.

**Figure 4 ijms-24-12811-f004:**
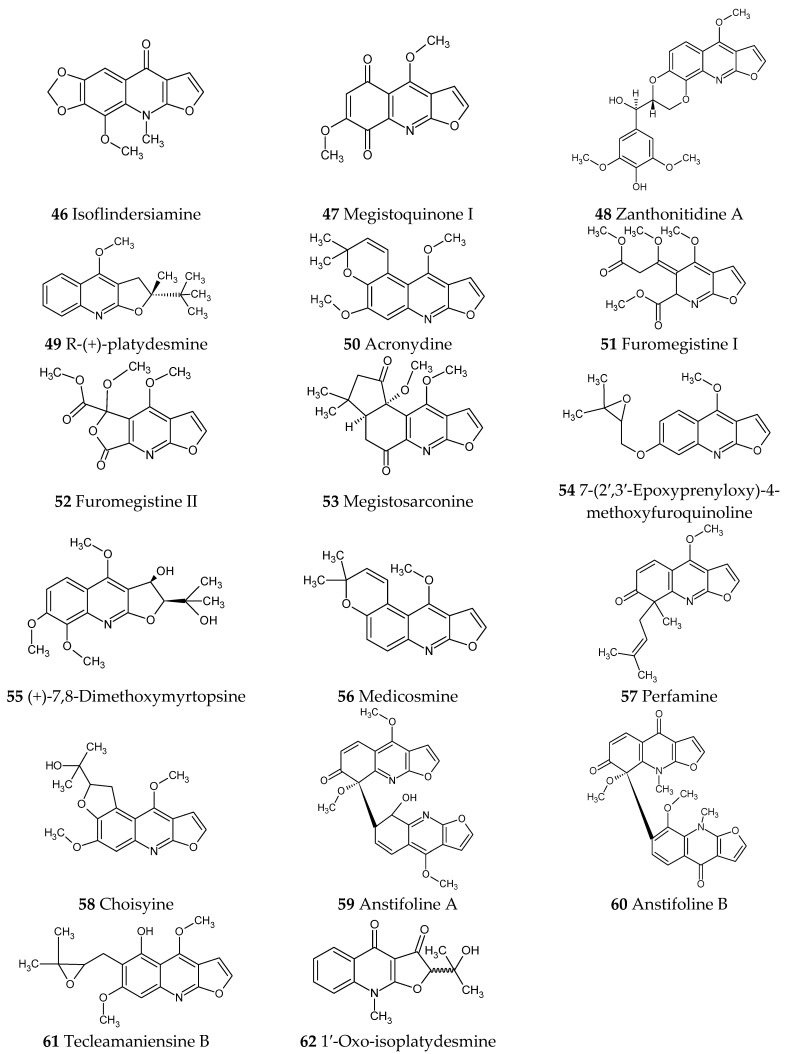
Chemical structures of compounds **46**–**62**.

**Table 1 ijms-24-12811-t001:** Selected alkaloids with examples of plants in which furoquinoline alkaloids occur and their main biological activity.

No.	Compound	Source	Activity	References
**1**	Dictamnine	*Ruta* sp., *Dictamnus* sp., *Haplophyllum* sp., *Zanthoxylum* sp., *Esenbeckia* sp., *Euodia* sp., *Flindersia* sp., *Geijera* sp., *Teclea afzeli*, *Cajanus cajan*, *Toddalia asiatica*, *Sigmatanthus trifoliatus*	Antibacterial, Antiviral, Antifungal, Antiprotozoal, Anticancer, Anti-inflammatory, Antioxidant, Cardiovascular, Antiplatelet, Antiosteoporosis, Anti-anaphylactoid	[1,4,5,6,7,8,9,10]
**2**	Skimmianine	*Zanthoxylum nitidum*, *Esenbeckia leiocarpa*, *Ruta graveolens*, *Zanthoxylum avicennae*, *T. asiatica*, *Tetradium ruticarpum*, *Dictamnus dasycarpus*, *Monnieria trifolia*, *Esenbeckia febrifuga*, *Allophylus bucharicum*, *Haplophyllum griffithianum*, *Evodia lepta*, *Araliopsis soyauxii*, *Medicosma fareana*, *S. trifoliatus*	Antibacterial, Antiviral, Antifungal, Antiprotozoal, Anticancer, Anti-inflammatory, Antioxidant, Neuroprotective, Cardiovascular, Antiplatelet, 5-HT receptor inhibition, Estrogenic	[1,4,5,7,9,10,11,12,13,14]
**3**	γ-Fagarine	*Ruta* sp., *D. dasycarpus*, *M. trifolia*, *E. febrifuga*, *Haplophyllum bucharicum*, *Z. nitidum*, *T. asiatica*, *Clausena emarginata*, *S. trifoliatus*	Antibacterial, Antiviral, Antifungal, Antiprotozoal, Anticancer, Anti-inflammatory, Cardiovascular, Antiplatelet, Estrogenic, Hepatoprotective	[1,5,7,9,10,15]
**4**	4-Hydro-7-hydroxy-8-methoxyfuroquinoline	*Ammi majus*	Anticancer	[3]
**5**	4-Hydro-7-hydroxy-8-prenyloxyfuroquinoline	*A. majus*	Anticancer	[3]
**6**	Haplopine	*Helietta parvifolia*, *H. bucharicum*, *T. asiatica*	Antibacterial, Antiviral, Antiprotozoal, Anticancer, Anti-inflammatory, Cardiovascular, Antiplatelet, Estrogenic	[5,9,16]
**7**	Isohaplopine	*Teclea simplicifolia*	Antibacterial, Antifungal	[1]
**8**	Kokusaginine	*Balfourodendron riedelianum*, *Melicope bonickii*, *Haplophylum* sp., *E. leiocarpa*, *T. afzelii*, *Teclea nobilis*, *H. parvifolia*, *M. trifolia*, *E. febrifuga*, *A. soyauxii*, *Vepris* sp.	Antibacterial, Antiviral, Antifungal, Antifouling, Antiprotozoal, Anticancer, Neuroprotective, Cardiovascular, Antiplatelet, 5-HT receptor inhibition	[1,13,16,17,18]
**9**	Pteleine	*D. dasycarpus*	Antibacterial, Anticancer, Antiplatelet	[11]
**10**	Lecomtequinoline A	*Vepris lecomteana*	Antibacterial	[19]
**11**	Robustine	*D. dasycarpus*, *H. bucharicum*, *Z. nitidum*	Antibacterial, Antifungal, Antiprotozoal, Anticancer, Antioxidant, Cardiovascular, Antiplatelet	[5,7]
**12**	5-(1,1-Dimethylallyl)-8-hydroxyfuro[2-3-b] quinolone	*Ruta chalepensis*	Antifungal, Antiprotozoal	[1]
**13**	Acronycidine	*Teclea amaniensis*	Antiprotozoal	[1]
**14**	Confusadine	*Melicope semecarpifolia*	Antiplatelet	[20]
**15**	Leptanoine C	*E. lepta*	Antiprotozoal	[12]
**16**	Heliparvifoline	*Melicope madagascariensis*	Antiprotozoal	[21]
**17**	Haplopine-3,3′-dimethylallyl ether	*Melicope moluccana*	Antiprotozoal, Antioxidant	[22]
**18**	Evolitrine	*D. dasycarpus*, *S. trifoliatus*	Anticancer, Anti-inflammatory, Neuroprotective, Cardiovascular, Antiplatelet	[10,11]
**19**	Maculosidine	*Vepris uguenensis*, *Ptelea trifoliata*	Anticancer	[23,24]
**20**	7-(2′-Hydroxy-3′-chloroprenyloxy)-4-methoxyfuroquinoline	*Melicope* sp.	Anticancer	[20]
**21**	4,5,6,7,8-Pentamethoxyfuroquinoline	*Vepris punctate*	Anticancer	[20]
**22**	5-Methoxyrobustine	*Dictamnus angustifolius*	Anticancer	[25]
**23**	Aegelbine A	*Aegle marmelos*	Anticancer	[26]
**24**	Aegelbine B	*A. marmelos*	Anticancer	[26]
**25**	7-Isopentenyloxy-γ-fagarine	*Haplophyllum ferganicum*, *H. latifolium*, *H. perforatum*	Anticancer	[24]
**26**	Confusameline	*M. semecarpifolia*	Anticancer, Cardiovascular, Antiplatelet, 5-HT receptor inhibition	[27]
**27**	Clausenaside F	*Clausena lansium*	Antioxidant	[26]
**28**	Melineurine	*E. lepta*	Neuroprotective	[12]
**29**	Leptanoine A	*E. lepta*	Neuroprotective	[12]
**30**	Leptanoine B	*E. lepta*	Neuroprotective	[12]
**31**	Tecleamaniensine A	*Teclea amanuensis*	Anti-insect	[1]
**32**	4-Hydroxyl-7,8-dimethoxy-furoquinoline	*Z. nitidum*	-	[7]
**33**	Evoxine	*Teclea gerrardii*	Antiprotozoal	[20]
**34**	Nkolbisine	*T. nobilis*, *M. trifolia*	Antibacterial, Antiparasitic	[28,29]
**35**	Flindersiamine	*B. riedelianum*, *H. parvifolia*, *Vepris* sp. (without *V. glomerata*), *E. febrifuga*, *A. soyauxii*	Antibacterial, Antifungal, Antifouling, Antiprotozoal, Anticancer	[1,13,16,17]
**36**	Maculine	*T. nobilis*, *Vepris* sp. (without *V. glomerata*), *A. soyauxii*	Antibacterial, Antifungal, Antiprotozoal, Anticancer	[13,17]
**37**	5-Methoxymaculine	*V. punctate*	Anticancer	[20]
**38**	5,8-Dimethoxymaculine	*V. punctate*	Anticancer	[20]
**39**	Isodictamnine	*Glycosmis arborea*	Anticancer, Antiplatelet	[20]
**40**	Iso-γ-fagarine	*D. angustifolius*	Anticancer, Anti-inflammatory	[25]
**41**	Isopteleine	*D. angustifolius*, *D. caucasicus*	Anticancer, Antiplatelet	[24]
**42**	Dictangustine-A	*D. dasycarpus*	Anti-inflammatory	[26]
**43**	Isomaculosidine	*D. angustifolius*, *D. caucasicus*, *D. dasycarpus*	Anti-inflammatory	[24,26]
**44**	8-Hydroxy-9-methyl-furo[2,3-b]quinolin-4(9H)-one	*D. angustifolius*	Antiplatelet	[26]
**45**	Isokokusaginine	*R. chalepensis*	-	[30]
**46**	Isoflindersiamine	*H. parvifolia*	-	[16]
**47**	Megistoquinone I	*Sarcomelicope megistophylla*	Antibacterial	[20]
**48**	Zanthonitidine A	*Z. nitidum*	Antibacterial	[7]
**49**	R-(+)-platydesmine	*Z. nitidum*	Antiviral, Antifungal	[7]
**50**	Acronydine	*T. amaniensis*	Antiprotozoal	[1]
**51**	Furomegistine I	*S. megistophylla*	Anticancer	[20]
**52**	Furomegistine II	*S. megistophylla*	Anticancer	[20]
**53**	Megistosarconine	*S. megistophylla*	Anticancer	[20]
**54**	7-(2′,3′-Epoxyprenyloxy)-4-methoxyfuroquinoline	*Melicope* sp.	Anticancer	[20]
**55**	(+)-7,8-Dimethoxymyrtopsine	*Melicope* sp.	Anticancer	[20]
**56**	Medicosmine	*Boronella konambiensis*	Anticancer	[20]
**57**	Perfamine	*Haplophyllum acutifolium*, *H. perforatum*	Anticancer	[24]
**58**	Choisyine	*Choisya ternata*, *Choisya dumosa var. Arizonica*	Antinociceptive	[31]
**59**	Anstifoline A	*D. angustifolius*	Anticancer	[32]
**60**	Anstifoline B	*D. angustifolius*	Anticancer	[32]
**61**	Tecleamaniensine B	*T. amanuensis*	Anti-insect	[1]
**62**	1′-Oxo-isoplatydesmine	*D. dasycarpus*	-	[11]

## Data Availability

Data sharing not applicable.

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
