# Peer review of "Furoquinoline Alkaloids: Insights into Chemistry, Occurrence, and Biological Properties"

_ijms, 2023, doi:10.3390/ijms241612811_

Round 1
Reviewer 1 Report
The manuscript ID ijms-2510266 compiles information about a small set of furoquinoline alkaloids and a compendium of biological properties. The manuscript has interesting elements. However, several issues limit its content and presentation to being considered further, and, in my opinion, the content and type of compounds are not enough to be considered for a review in this high-impact journal. In addition, the manuscript has several stylistic and grammar mistakes, many typos, and its presentation is not carefully outlined. For instance, the citation style is rare. The chemistry view of this compilation only involved 28 alkaloids, and the rest of the manuscript is related to describing, as a catalog, the reported biological properties of some alkaloids. However, this information is not well presented, unorganized, and insufficient for some furoquinoline alkaloid representatives, whose biological properties should include more information and reports. Finally, the discussion is laconically developed, and the conclusions section summarizes information without conceptual meanings to be considered conclusive passages.
As mentioned, the manuscript has several stylistic and grammar issues to be considered further.
Author Response
Manuscript ID: IJMS-2510266
Responses to the Reviewer 1 comments:
Dear Reviewer,
We are greatly obliged for having received the Reviewers’ valuable opinion and helpful suggestion on our manuscript. The manuscript with corrections is attached. The correction was made using the track changes mode. Additionally, added fragments of text have been marked in yellow.
The replies to the specific comments are listed below.
The manuscript ID ijms-2510266 compiles information about a small set of furoquinoline alkaloids and a compendium of biological properties. The manuscript has interesting elements. However, several issues limit its content and presentation to being considered further, and, in my opinion, the content and type of compounds are not enough to be considered for a review in this high-impact journal.
Answer: Thank you very much for your thorough and constructive review of the manuscript. As suggested, we have significantly increased the number of compounds discussed.
In addition, the manuscript has several stylistic and grammar mistakes, many typos, and its presentation is not carefully outlined. For instance, the citation style is rare.
Answer: We apologize for the shortcomings. We have made corrections, the citation has been corrected, and the language correction has been made (Translmed Publishing Group).
The chemistry view of this compilation only involved 28 alkaloids, and the rest of the manuscript is related to describing, as a catalog, the reported biological properties of some alkaloids. However, this information is not well presented, unorganized, and insufficient for some furoquinoline alkaloid representatives, whose biological properties should include more information and reports. Finally, the discussion is laconically developed, and the conclusions section summarizes information without conceptual meanings to be considered conclusive passages.
Answer: Thank you for all valuable suggestions. As suggested, we have increased the number of compounds discussed. We have decided to remove Table 2 so that it does not feel like reading a catalogue. We hope this makes the publication more consistent. We have expanded the description of pharmacological actions. We modified the discussion and conclusion.
We hope that we improved the manuscript as suggested.
Please, accept my best regards,
Yours sincerely,
Agnieszka Szewczyk
Reviewer 2 Report
This paper provides a timely review of key biological properties of furoquinolines alkaloids. The article appears to be well-constructed and its coverage is thorough. The literature citations are adequate. The usual minor problems, e.g., punctuation (line 47: … cinnamic acid, such … à … cinnamic acid. Such…), unnecessary sentence fragments (line 49: …ring, in the case of furoquinoline alkaloids, furan… à … ring, furan…), etc., can be resolved upon thorough proofreading; otherwise the paper is well-written and clear.
Acceptance is recommended.
Author Response
Manuscript ID: IJMS-2510266
Responses to the Reviewer 2 comments:
Dear Reviewer,
We are greatly obliged for having received the Reviewers’ valuable opinion and helpful suggestion on our manuscript. The manuscript with corrections is attached. The correction was made using the track changes mode. Additionally, added fragments of text have been marked in yellow.
The replies to the specific comments are listed below.
This paper provides a timely review of key biological properties of furoquinolines alkaloids. The article appears to be well-constructed and its coverage is thorough. The literature citations are adequate. The usual minor problems, e.g., punctuation (line 47: … cinnamic acid, such … à … cinnamic acid. Such…), unnecessary sentence fragments (line 49: …ring, in the case of furoquinoline alkaloids, furan… à … ring, furan…), etc., can be resolved upon thorough proofreading; otherwise the paper is well-written and clear.
Acceptance is recommended.
Answer: Thank you very much for the reviewer's kind opinion. We tried to correct all the shortcomings. The text has also been linguistically corrected (Translmed Publishing Group).
We hope that we improved the manuscript as suggested.
Please, accept my best regards,
Yours sincerely,
Agnieszka Szewczyk
Round 2
Reviewer 1 Report
The authors addressed my comments, and the manuscript improved in quality and content. Therefore, it can be considered further. However, I recommend formatting adequately Table 1, and the structure size of contained alkaloids must be homogenized.
Author Response
Responses to the Reviewer 1 comments (round 2):
Dear Reviewer,
We are greatly obliged for having received the Reviewers’ valuable opinion and helpful suggestion on our manuscript. The manuscript with corrections is attached. Changes are marked in yellow.
The replies to the specific comments are listed below.
“The authors addressed my comments, and the manuscript improved in quality and content. Therefore, it can be considered further. However, I recommend formatting adequately Table 1, and the structure size of contained alkaloids must be homogenized.”
Answer: Thank you for your valuable suggestions. Table 1 has been corrected and formatted. As suggested by the Academic Editor, the chemical structures have been collected in the form of four figures. We hope that the corrections will make the manuscript more legible.
We hope that we improved the manuscript as suggested.
Please, accept my best regards,
Yours sincerely,
Agnieszka Szewczyk
